# A Tool for High-Resolution Volumetric Optical Coherence Tomography by Compounding Radial-and Linear Acquired B-Scans Using Registration

**DOI:** 10.3390/s22031135

**Published:** 2022-02-02

**Authors:** Christian M. Bosch, Carmen Baumann, Shervin Dehghani, Michael Sommersperger, Navid Johannigmann-Malek, Katharina Kirchmair, Mathias Maier, Mohammad Ali Nasseri

**Affiliations:** 1Computer Aided Medical Procedures and Augmented Reality, Technical University Munich, 85748 Munich, Germany; shervin.dehghani@tum.de (S.D.); michael.sommersperger@tum.de (M.S.); 2Augenklinik und Poliklinik, Klinikum Rechts der Lsar der Technische Universität München, 81675 München, Germany; carmen.baumann@mri.tum.de (C.B.); Navid.Johannigmann@mri.tum.de (N.J.-M.); Katharina.Kirchmair@mri.tum.de (K.K.); Mathias.Maier@mri.tum.de (M.M.); ali.nasseri@mri.tum.de (M.A.N.)

**Keywords:** optical coherence tomography, OCT, high-resolution, 3D OCT, volumetric OCT, ICP, inflection point, macular hole

## Abstract

Optical coherence tomography (OCT) is a medical imaging modality that is commonly used to diagnose retinal diseases. In recent years, linear and radial scanning patterns have been proposed to acquire three-dimensional OCT data. These patterns show differences in A-scan acquisition density across the generated volumes, and thus differ in their suitability for the diagnosis of retinal diseases. While radial OCT volumes exhibit a higher A-scan sampling rate around the scan center, linear scans contain more information in the peripheral scan areas. In this paper, we propose a method to combine a linearly and radially acquired OCT volume to generate a single compound volume, which merges the advantages of both scanning patterns to increase the information that can be gained from the three-dimensional OCT data. We initially generate 3D point clouds of the linearly and radially acquired OCT volumes and use an Iterative Closest Point (ICP) variant to register both volumes. After registration, the compound volume is created by selectively exploiting linear and radial scanning data, depending on the A-scan density of the individual scans. Fusing regions from both volumes with respect to their local A-scan sampling density, we achieve improved overall anatomical OCT information in a high-resolution compound volume. We demonstrate our method on linear and radial OCT volumes for the visualization and analysis of macular holes and the surrounding anatomical structures.

## 1. Introduction

Since its invention in the early 1990s, optical coherence tomography (OCT) has mainly been utilized in ophthalmology [1,2,3], and, with the introduction of spectral domain optical coherence tomography (SD-OCT) in the late 2000s, it has advanced to one of the main diagnostic tools, particularly for diseases affecting the central retina (macula) [4,5,6]. Due to limitations in high-density B-scan slice acquisition and wavelength, as well as the need for sophisticated reconstruction techniques to improve image quality, linear volumetric OCT has remained the standard for volumetric reconstruction [7].

In most clinical settings, the OCT systems provide two different modes of scanning patterns. The more commonly used linear pattern consists of contiguous parallel B-scans, as shown in image (A) of Figure 1, whereas the radial scanning pattern is composed of B-scans, which are acquired with a uniform angular distance around a mutual center point, as shown in image (B) of Figure 1. Linearly acquired B-scans use a scanning protocol with uniform spacing between the parallel B-scan slices. Due to the angular relationship of the B-scans in the radial scanning pattern, the A-scan sampling density of radial volumes is highest at the scan center and decreases with increasing distance from the center.

While the linear mode enables the imaging and evaluation of conditions, which affect larger anatomical areas, such as retinal thickening in diabetic macular edema or changes in age-related macular degeneration, the radial scan mode is superior in the diagnosis of conditions such as macular holes, which affect only a very small area confined to the foveal center. Since macular holes have been shown to be significantly asymmetric [8,9], the radially arranged B-scans allow for a more detailed evaluation of the architecture and the dimension of macular holes, considering all meridians or axes equally [8]. Additionally, the asymmetric restoration process of retinal layers, following successful macular hole surgery, can be analyzed precisely using radially acquired OCT B-scans [10]. Thus, radial OCT volumes are highly effective to analyze macular holes, forming the basis of many medical and surgical treatment decisions.

For the precise visualization and analysis of anatomical structures and pathologies, there is a great need for automated volumetric 3D OCT scanning, allowing clinicians to conduct an objective analysis of relevant pathologies in their three-dimensional aspects, 360 degrees, which could improve not only the understanding of diseases but also the precision and reliability of measurements that are crucial for treatment planning, prognostic guidance and research in various fields. We aim to fuse both scans to achieve higher-resolution A-scans, which enhance the volumetric interpretation of lateral resolution, as similarly discussed in [11].

In recent years, works have been presented to explore image enhancement strategies using interpolation or pattern recognition [12]. However, such approaches are mostly limited to technological factors such as the range of wavelengths, which, in turn, restrict the OCT B-scan quality, and thus the possibility of applying image enhancement in the post-processing stage [13].

Recent improvements in OCT imaging quality have been achieved by applying deep learning models to create synthesized scans to improve the signal-to-noise ratio (SNR) of time-domain (TD) OCT B-scans to spectral-domain (SD) OCT B-scans [14]. In [15], the potential power of radial scanning patterns is discussed; however, to the best of our knowledge, there has not been any work regarding the compounding radial and linear OCT volumes in recent years.

To improve OCT image quality, the maximum number of radial B-scans is selected, providing the highest possible density of scans around the chosen center point. The maximum available slices, in this case, are 48 radial B-scans, taken continuously over 360° degrees. Due to the radial overlapping of all 48 B-scans, a high-density volumetric quality is achieved around the center. This enables the clinician to look at any of the 48 cross-sectional B-scans of the retina, which is not possible in a regular linear volumetric OCT setup.

To enhance the information in less densely sampled areas further from the center point, our approach consists of the initial registration of the radial and linear OCT volumes. The registration allows for gaps in the radial volume to be filled in with information from the corresponding linear registered volume, permitting the acquisition of continuous data from focally confined structures or lesions, as well as of the morphology or pathologies affecting wider areas of the retina.

Therefore, in this paper, we propose a novel technique to enhance the image quality and usability of volumetric OCT by using a variation of the Iterative Closest Point (ICP) algorithm to combine the advantages of radial and linear B-scans.

The processing of the radial and linear volumetric OCT scans is applied offline; therefore, the acquisition time of the volumetric OCT scans is not prolonged. Our algorithm compensates for motion between linear and radial scan acquisitions, in case of possible movements between the two scans.

## 2. Methods

In the following section, we present the steps, illustrated in Figure 2, that were employed to obtain radial and linear volumes, which were registered and compounded step by step.

### 2.1. Linear Volumetric Optical Coherence Tomography

To create a linear volumetric OCT, the center point around which the scanning was performed (pattern center) was positioned on the pathology of interest. A total of 145 high-resolution (1024 × 310 pixels) B-scans were taken, with a high-density scanning protocol of 30 µm between each adjacent B-scan, which is approximately 6 times denser than the standard linear scanning protocol, with 200 µm line spacing between adjacent B-scans. Using the high-resolution, high-density linear scan pattern resulted in a linear volumetric OCT with superior quality, which ultimately led to further structural information, making it more likely to include all pathological information. The acquisition time of the high-resolution, high-density linear B-scans was about 10 s (Spectralis OCT, Heidelberg Engineering). Each B-scan was then consecutively linear-aligned with respect to the horizontal spacing and converted to a volumetric linear OCT.

### 2.2. Radial Volumetric Optical Coherence Tomography

Similar to the linear volumetric OCT, a radial volumetric OCT was created by positioning the pattern center onto the pathology of interest. The ability of choosing the volumetric image center allows for a decent volumetric OCT in the neighbourhood of 1000 µm around the pattern center due to the high sampling rate. The radial scans were acquired with an angular interval of 3.75° between adjacent B-scans. For a full radial OCT scan, an interval of 180 should be covered by the device, which means that the space is represented by $\frac{180{}^{\circ}}{3.75{}^{\circ}}=48$ individual B-scans, as shown in image (A) of Figure 1. The acquisition time of one full radial volumetric OCT scans was about 10 s (Spectralis OCT, Heidelberg Engineering). Each of the obtained B-scans was then placed in its respective position, from 0 to 180, in a 3D space. After all B-scan slices were properly collected, we ran a modified volume-compounding algorithm on all 48 slices to create a radial volumetric OCT. The modified volume-compounding of the 48 radial slices to a radial volumetric OCT took, on average, 12 s.

### 2.3. Iterative Closest Point

After retrieving the two different OCT volumes from the previous step, it is necessary to register the two volumes to align them in case of further processing. This problem is a classic problem in the computer vision community, and the conventional algorithm was ICP [16], designed to calculate the transformation T(AB) that aligns point cloud *A* with point cloud *B*.

The point-wise ICP first proposed by Chen and Medioni, and Besl and McKay, is an algorithm used to minimize the difference between two point clouds [17,18]. The algorithm consists of three steps: selecting corresponding point cloud pairs, calculating the rigid body transformation for one point cloud, and applying this transformation. These steps are repeated until the algorithm converges or a certain threshold is reached. For each iteration, the algorithm provides a rotation and transformation matrix. These matrices are then multiplied to each point of the moving point cloud. The ICP computation time takes about 25–30 s. To assist the algorithm in finding valid pairs, converging faster and improving its accuracy, we brought both point cloud surfaces into the same plane.

Having two point clouds with *S* as the static point cloud (Figure 3A) and *M* as the moving point cloud (Figure 3B), where $S_{i}$ and $M_{i}$ represent point sets of the respective point cloud, a transformation process between *S* and *M* is required, which minimizes the difference between these two point clouds. Therefore, the distance between each point-wise pair is calculated using the Euclidean distance until a certain threshold is reached.
(1)$$\begin{array}{c} dist=\min\sqrt{M_{i}^{2}-S_{i}^{2}} \end{array}$$

Firstly, we calculated the alignment between *S* and *M* by computing the rotation *R* and transformation *t* matrix. Each point set of the moving point cloud $M_{i}$ was multiplied with *R* and *t* and then the error between each point pair was calculated using the Euclidean distance. The aim was to minimize the error E:(2)$$\begin{array}{c} E\left(R,t\right)=\sum_{i=1}^{N}{∥M_{i}-\left(S_{i}*R+t\right)∥}^{2} \end{array}$$

To minimize *E* in Equation (Equation 2) we applied the Singular Value Decomposition (SVD). A prerequisite for computing the SVD is the centroid alignment of both point sets $C_{S}$ and $C_{M}$, followed by the alignment of all points in *S* and *M*. Upon completion of these, we computed the covariance matrix
(3)$$\begin{array}{c} H=M*S^{T} \end{array}$$
and make the SVD as
(4)$$\begin{array}{c} H=U\Sigma V^{T} \end{array}$$
where U and V are orthogonal matrices and $\Sigma =diag(\sigma_{1},\sigma_{2},...,\sigma_{d})$ such that $\sigma_{1}\geq \sigma_{2}\geq ...\geq \sigma_{d}\geq 0.$ Following this, it can be proved that, to minimize Equation (Equation 2), the rotation matrix should be defined as
(5)$$\begin{array}{c} R=V*U^{T} \end{array}$$
and the translation vector following
(6)$$\begin{array}{c} t=C_{S}-R*C_{M} \end{array}$$
where $C_{S}$ and $C_{M}$ represent the respective centroids of the point sets *S* and *M*. Each iteration of the ICP consists of the two most important elements of the algorithm: the rotation matrix *R* and the translation vector *t*. At the beginning of each iteration, we computed the error E and if the error was larger than a preset threshold, the algorithm continued to iterate; otherwise, the algorithm stopped and the ICP had reached its optimal solution.

Once the ICP algorithm was completed, as seen in Figure 4, and the transformation and rotation from the static to the moving point cloud was known, we created a mask of the static point cloud (radial point cloud). This mask was applied and subtracted from the registered moving point cloud (linear point cloud). The static point cloud was then fitted with the new moving point cloud to achieve a gap-less and final point cloud. While the radial B-scans slices provided the highest possible resolution in the immediate proximity of the acquisition center, information gaps appeared in the periphery of the volume, which are now being filled with the linear volumetric information. This results in the highest possible resolution by combining the radial and linear volumes, and without averaging or interpolating any data.

### 2.4. Inflection Point & Compounding

At this stage, we had the registered volumes from both scanning patterns, with the A-scan view illustrated in Figure 5. We presumed that the A-scan density of a radial scanning pattern is higher closer to the center of the scans and starts to decrease as we move further peripherally. Meanwhile, we assumed the inverse for the linear scanning pattern. Hence:(7)$$\begin{array}{cc} d_{radial}\left(x\right):= & \frac{\left|\mathrm{Radial}A-\mathrm{scans}\mathrm{with}\mathrm{distance}x\mathrm{to}\mathrm{center}\right|}{\pi x^{2}}\\ d_{linear}\left(x\right):= & \frac{\left|\mathrm{Linear}A-\mathrm{scans}\mathrm{with}\mathrm{distance}x\mathrm{to}\mathrm{center}\right|}{\pi x^{2}}\\ t(x):= & \frac{d_{linear}\left(x\right)}{d_{radial}\left(x\right)} \end{array}$$
where $d_{radial}\left(x\right)$ is the density function of A-scans from the radial scanning pattern at a distance of *x* from the center, $d_{linear}\left(x\right)$ the density function of A-scans from the linear scanning pattern at distance *x* from the center and t(x) the ratio of $d_{linear}\left(x\right)$ to $d_{radial}\left(x\right)$. We named $d_{M}\left(x\right)$ spatial A-scan resolution of the M scanning pattern, at distance *x* from the center.

Since $d_{radial}\left(x\right)$ is a strictly decreasing function and $d_{linear}\left(x\right)$ is strictly increasing function, a value *p* should exist, for which condition t(p)=1 is fulfilled. For all radial masks with *r* in [0,p], the radial scanning pattern had a higher sampling density. Consequently, the same held true for the linear scanning pattern in case of r>p. In the following, we called the point *p* the Inflection Point, according to which we compounded the A-scans of two volumes using
$$F\left(x\right)=\left\{\begin{array}{cc} R(x), & \mathrm{if}\;dist(x)\leq p\\ L(x), & \mathrm{otherwise}, \end{array}\right.$$
where $x\in R^{2}$ is the projected 2D point of an A-scan, as shown in Figure 5 , R(x) the A-scan of *x* obtained from radial pattern, with L(x) being the same but obtained from the linear pattern, and F(x) is the fused A-scan of *x*, while dist(x) is the distance of *x* from the center and *p* is the Inflection Point.

The existence and value of the inflection point fully depends on the characteristics of the OCT device. The A-scan densities and the B-scan intervals play a crucial role in this analysis. For the purpose of this work, our image acquisitions were made using Spectralis OCT, Heidelberg Engineering. This led to an A-scan density of 1 scan per 6.34 µm for both linear and radial scanning patterns. The B-scans were taken at an interval of 30 µm for the linear pattern (which led to 145 B-scans) and 3.75 degrees for the radial pattern (which led to 48 scans). By analysing the A-scan density of both patterns in radial masks with a variable radius (Figure 6), we could verify the existence of the inflection point, with R=1358.3 µm.

### 2.5. Enhancement of Spatial A-Scan Resolution

The enhancement of the spatial A-scan resolution in this work is proposed by compounding radial and linear OCT B-scans.

As illustrated in Figure 5, the key measure of this study is the spatial A-scan resolution. This value is higher for the radial scan until the inflection point, and for the linear scan afterwards. There is more information in the radial OCT B-scans and this is, therefore, preferred until the inflection point at 1358.3 µm, and the linear OCT B-scans afterwards.

## 3. Experiments

We used a Heidelberg Engineering Spectralis OCT system, which is capable of penetrating through the retinal pigment epithelium (RPE) and, therefore, enables imaging of the choroid, with a wavelength of 870 nm. As a result, we are able to achieve a lateral resolution of 5.54 µm/pixel and an in-depth resolution of 3.87 µm/pixel.

During the acquisition, motion artifacts rarely occurred, due to the build in the real-time eye-tracking system in the Spectralis OCT, Heidelberg Engineering. The real-time eye-tracking system stops/cancels the scan as soon as eye-movement of the patient is captured, thus preventing motion artifacts in the OCT output.

All our methods ewre developed in python and visualized using the graphical user interface (GUI) ImFusion Suite. All the computations were performed on an AMD Ryzen 7 4800H CPU.

## 4. Results

This section presents the results of the mentioned radial volumetric OCT, the radial and linear fused volumetric OCT. We also discuss the importance of the inflection point, analytically prove the existence of the point and calculate its value according to our setup.

### 4.1. Radial Volumetric OCT Reconstruction

In Figure 7, two images of the radial volumetric reconstruction are displayed. Both images show the same macular hole. A total of 48 radially acquired B-scans with 1024 × 312 pixels were compounded into a volumetric OCT with 1024 × 312 × 1024 pixels. The approximate center of the macular pathology was centered near the pattern center. This enhances the illustration of the volumetric OCT due to the radial acquisition technique and the circular nature of the pathology. These two properties, when combined, allow for a high scan density, and thus a detailed image quality of the macular hole itself and its surrounding (Figure 7A), which allows clinicians the most accurate evaluation of the macular hole´s architecture and precise measurement of the relevant dimensions on which treatment decisions are based. The volumetric reconstructed OCT is viewable from different perspectives (Figure 7B), providing further information on exterior features such as surface deformation near the volumetric center.

### 4.2. Detectability of the Asymmetry of Macular Holes due to 3D reconstruction

Figure 8 displays three close-up images of the same macular hole at the level of different layers. At each level, clinicians can extract the desired information about the macular hole’s configuration and dimensions such as the widest surface opening diameter, minimum linear diameter and base diameter, and the corresponding areas. As illustrated in Figure 8 row 2, macular holes are not necessarily perfectly circular, as one could assume from the B-scan view in row 1. Therefore, radial volumetric reconstruction provides superior information compared to linear alone in terms of the shape, size and structural analysis of macular holes.

### 4.3. Comparison of Radial and Fused OCT Reconstruction

Figure 9 shows a comparison of a macular hole pathology reconstruction with the two different major techniques outlined in this paper: radial volumetric OCT reconstruction (row 1/blue frame) and radial- and linear volumetric fused OCT reconstruction (row 2/yellow frame). The blue and yellow outer frame correspond to the cross-section from which the B-scan was taken, visualized by the line in column 3. Images in row 1; column 1 and column 2 are acquired at 1024 × 312 pixels and present two randomly chosen cross-sectional slices of the radial volumetric OCT reconstruction at different heights within the volume, in proximity of the macular hole. Displayed in row 2; column 1 and column 2 are two cross-sectional slices of the radial and linear fused volumetric OCT. The radial image data size is 1024 × 312 pixels. The linear acquired B-scan images have a size of 1024 × 310 pixels, and the linear reconstructed OCT volume has 1024 × 310 × 838 pixels, whereas the radial reconstructed OCT volume has a size of 1024 × 312 × 1024 pixels. The images in row 2 demonstrate the benefit of the radial and linear fused OCT reconstruction, as compared to the conventional radial reconstructed OCT with 48 slices in row 1.

### 4.4. Comparison of Individual B-Scan Slice

Figure 10 shows three cross-sectional B-scan slices of the same cut plane from three different OCT volumes. (A) A cross-sectional B-scan slice of the radially reconstructed volumetric OCT and (B) a cross-sectional B-scan slice of the linear reconstructed volumetric OCT, demonstrating the strengths and weaknesses of both. In image (A), close to the acquisition center, a high density of structural information is achieved due to the overlap of all 48 B-scans. However, with increasing distance from the pattern center, due to the nature of the radial acquisition, gaps in the B-scan slice appear and increase in size, representing a loss of structural information (area (2) and (3)). As our viewpoint is parallel throughout the entire volume, visible gaps occur due, as each B-scan slice is rotated by 3.75 degrees around the B-scan rotation axis. Hence, the shown cross-section in Figure 9A has gaps at (2) and (3) (black areas). This can be observed, as seen in Figure 9 row 1. The further away from the center point, the more gaps with missing information occur. In contrast, the linear cross-sectional B-scan slice (B) provides less detailed information around the center of the OCT scan pattern, but delivers continuous structural information across the entire image range, making it the preferred scan mode for pathologies affecting wider areas of the macula. However, for focal lesions or pathologies such as a macular hole, the area of interest is very narrow, and detailed information on the immediate peri-focal region at the micrometer-level is crucial for the precise analysis of the architectural aspects of lesions and their dimensions in a 360° fashion. This is compromised in linearly acquired and linear volumetric reconstructed OCT B-scans due to their voxel size and linear spacing during acquisition, and radially obtained data with 48 overlapping B-scan slices close to the acquisition center provide further morphological details. Therefore, both image acquisition modes have advantages for different purposes. Image (C) shows the combined advantages of both radially and linear acquired data to achieve an improved B-scan slice providing superior image resolution.

### 4.5. Inflection Point

The inflection point, introduced in Section 2.4, is depicted from a aerial view in Figure 11. Figure 11A visualises a numerically calculated plot of the inflection point with exact pagination. Near the image center, the radial scanning pattern is represented by blue lines. Beyond the inflection point, the point at which the $d_{linear}$ is superior to the $d_{radial}$ is represented in tan. In Figure 11B, this scheme is visualized by our radial and linear fused reconstruction by selectively choosing the radial pattern until the inflection point at radius 1358.3 µm, and then the linear pattern beyond the inflection point. In Figure 11B, the radial scanning pattern is represented in red and the linear scanning pattern in grey. Image (B) nicely shows the asymmetry of the macular hole, as introduced in Figure 8.

We have applied our method for the analysis of macular holes, which have been shown to occupy an anatomical area of 212–1073 µm [19] in diameter. Accounting for the inflection point of the A-scan densities of radial and linear scans, it could be noticed that, around a radius of 1358.3 µm around the volume center, which is more then twice the radius of common macular holes [19], the radial scan has a higher spatial A-scan resolution. When positioning the volume center at the center of the macular hole, we could use our proposed method to maximize the amount of available information by using both the radial and linear OCT scans, as shown in Figure 6.

The inflection point value could be further increased by reducing the B-scan interval for the radial scanning pattern. Having an OCT device, which can obtain radial scans at a lower interval (which results in a higher number of B-scans), would result in a different outcome. For instance, if we can decrease the B-scan interval of the radial pattern to 1.5 degrees, to obtain a similar number of B-scans with a linear scan, (resulting 120 B-scans, which is still lower than the 145 B-scans of the linear scan), the inflection point value would be further increased to 3338 µm. However, due to the hardware limitations, this value could not be acquired, so there should be a trade-off between the linear and radial scanning patterns.

In the current clinical ophthalmic environment, a 3D visualization, as seen in Figure 11, is not accessible. Therefore, in Figure 12, we present the inflection point on the basis of an OCT B-scan. On the left-hand side of the OCT B-scan, the fundus view is present, with a yellow circle corresponding to the size of the inflection point. On the right-hand side, the area below the inflection point is marked with two solid yellow lines. Here, it is also notable that, in the case of a macular hole, the inflection point covers the entire pathology.

## 5. Discussion & Future Work

In this work, we presented a novel framework to assist ophthalmologists in their decision-making, patient consultation, treatment-planning and analysis. Our methods take advantage of the individual characteristics of radial- and linear volumetric OCT scans. Linear as well as radial OCT B-scans have been used in different contexts but never together. Radial-acquired B-scans allow clinicians to extract information from any orientation, whereas extracting the same information from the linear-acquired B-scans is difficult and often not possible [8]. The linear scanning pattern is more frequently used, as it enables clinician to evaluate more widespread conditions affecting larger areas. However, in less sampled regions further away from the acquisition center, due to the limitation in radial slices, crucial information is lost. Figure 8 shows a close-up of the density around the scan center, whereas Figure 7 shows the entire scan area, where the less-sampled areas are visible. In areas closer to the acquisition center, where the most significant information occurs, we chose the radial acquired B-scan slices with a high-density sampling rate. In less densely sampled areas further away from the acquisition center, we took advantage of the high-resolution high-density linear scans and created a resulting volume with the maximum density and maximum information of both scanning patterns. The improved image quality obtained by registering both scanning patterns together is clearly visible in Figure 10C. Even if linear B-scans with decreased intervals are required, e.g., 8 µm less per interval, compared with the current radial scanning pattern, there would still be an inflection point with *R* = 675.6 µm, which still covers the outer area of the macular holes. Figure 9 (column 3) additionally demonstrates the improvement we achieved by utilizing this method. The resolution and, therefore, anatomical information, is clearly finer in row 2 than in row 1. The resulting 3D model can be used to analyze morphological features and pathologies in a 3D manner, rather than relying on simple length measurements from the data in each plane. With this methodology, we obtain a base for future generic measurements and calculations. Automated measuring of the minimum linear diameter (MLD) and base diameter (BD), as well as volumetric modeling of the macular hole, remain a subject for future work. We aim to expand and modify the tool for diseases such as macular degeneration and age-related macular degeneration (AMD), neovascularization (NV) or retinal tear/detachment. This will enable clinicians to use a multi-purpose tool for ophthalmic diagnostics, assisting them in the application of 3D analysis to any pathology in a 360° space.

## 6. Conclusions

Poor image quality is a challenge in the field of ophthalmology at present. With this tool, we offer a solution by combining two available acquisition techniques to increase quality where needed. The entire process from image acquisition to post-processing and radial and linear volumetric fusion does not take longer than 70 s. With quantitative and qualitative tests, we evaluated our approach and were able to set an inflection point until the image quality of radial volumetric OCT is higher than linear volumetric OCT. This inflection point, as well as the related image quality, are limited by present-day technical components within the OCT devices. Looking forward, we expect these limitations to be overcome and, with this, the poor image quality of OCT. In case of such an event, our approach of radial and linear fusion would be further enhanced. A comparison of radial, linear, and radial and linear registered volumetric OCTs demonstrates the substantial information gain achieved by radial and linear volumetric OCT registration. We have proposed a methodology in which the clinicians can decide whether to use the radial or linear volumetric OCT itself, or to register them together. This allows for high-quality imaging in specific chosen areas and, furthermore, leads to a better understanding of certain pathologies, where only radial or linear A-scans or B-scans are insufficient. The high-resolution tool we present has the potential to support decision-making and help clinicians make further analysis easier and more accessible.

## Figures and Tables

**Figure 1 sensors-22-01135-f001:**
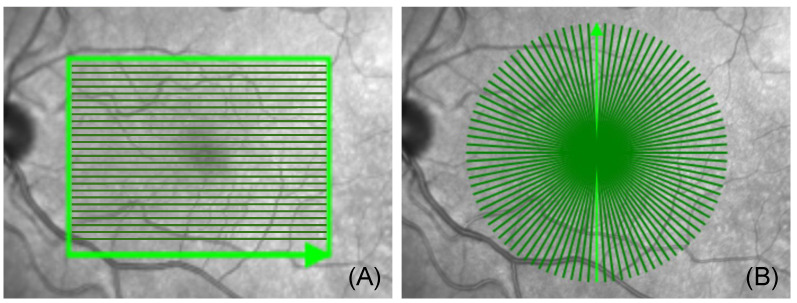
(**A**) shows the linear acquisition pattern, consisting of 145 linear slices around a center point with the first (1/145) acquisition slices highlighted in bright green. (**B**) shows the radial acquisition pattern consisting of 48 radians/slices around a center point, with the first (1/48) acquisition slice highlighted in bright green (Spectralis OCT, Heidelberg Engineering, Germany). Note the dense acquisition area around the acquisition center in image (**B**).

**Figure 2 sensors-22-01135-f002:**
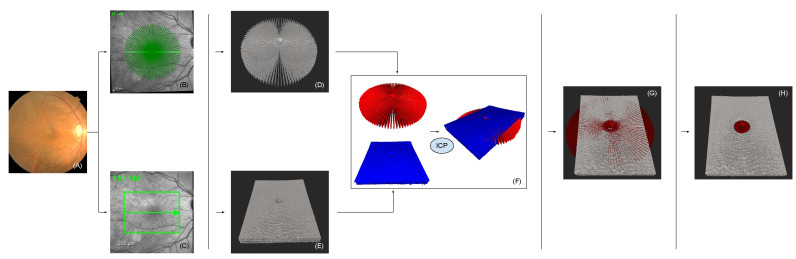
This figure reflects an overview of the entire pipeline used to accomplish radial and linear volumetric OCT registration and fusion. (**A**) shows the fundus image of the retina. Image (**B**,**C**) show the chosen scan area with the correlating scan pattern ((**A**) radial scan pattern; (**B**) linear scan pattern). The acquisition was subsequently performed, but post-processing was conducted simultaneously. Images (**B**,**D**) represent the 3D-OCT volumes that correspond to (**B**,**C**). (**F**) shows the point clouds generated from (**D**,**E**) and registered via ICP. Image (**G**) shows the radial volumetric OCT (red) fused with the linear volumetric OCT (gray). (**H**) presents the final result, with the advantageous chosen radial volumetric OCT area in red, surrounded by the linear volumetric OCT.

**Figure 3 sensors-22-01135-f003:**
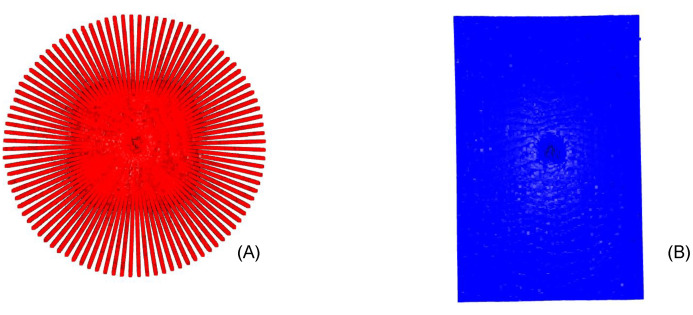
(**A**) presents the top view of the radial reconstructed point cloud. Note the high density of points in the area surrounding the image center. (**B**) shows the top view of the linear volumetric reconstruction point cloud.

**Figure 4 sensors-22-01135-f004:**
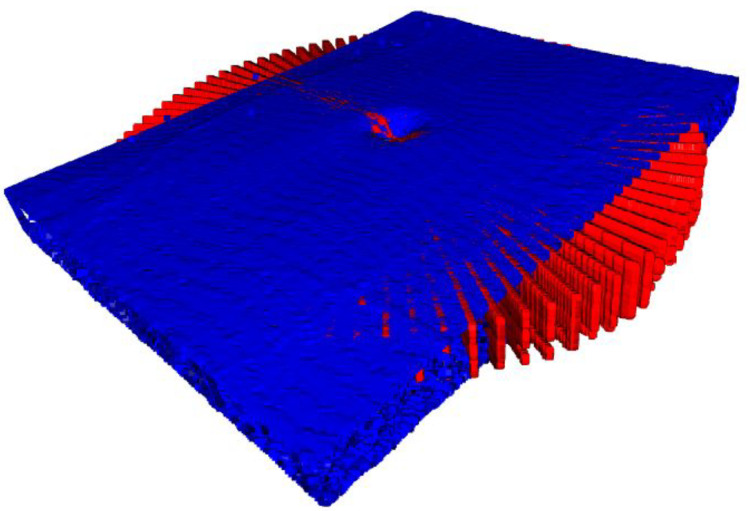
Visualized is a tilted and rotated view of the two point clouds after the ICP algorithm was completed, with the radial reconstructed point cloud as red and the linear reconstructed point cloud as blue volume.

**Figure 5 sensors-22-01135-f005:**
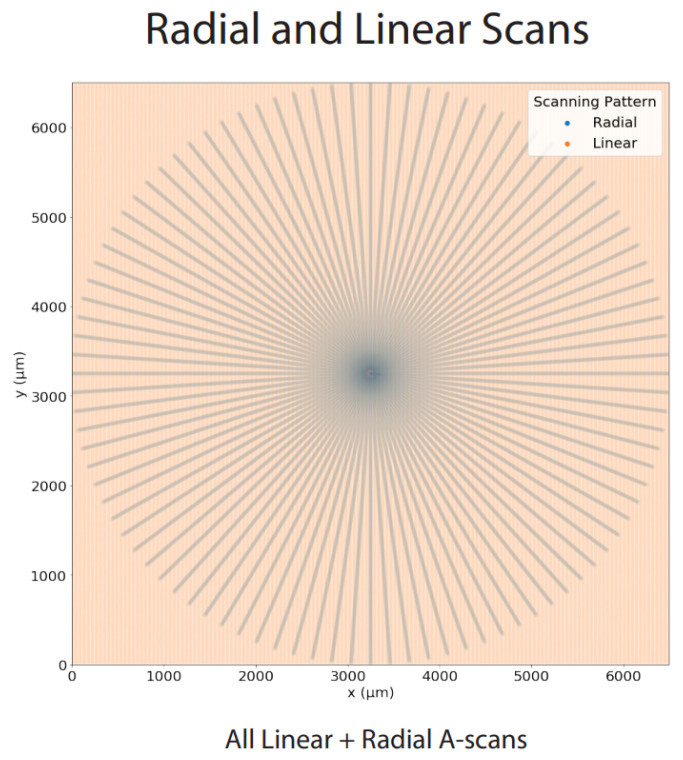
The registered A-scan view of both scanning patterns.

**Figure 6 sensors-22-01135-f006:**
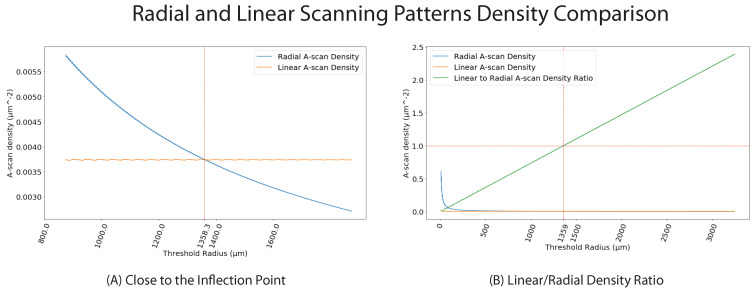
In plot (**A**) the intersection represents the existence of the Inflection Point. (**B**) shows the density ratio of linear to radial A-scans.

**Figure 7 sensors-22-01135-f007:**
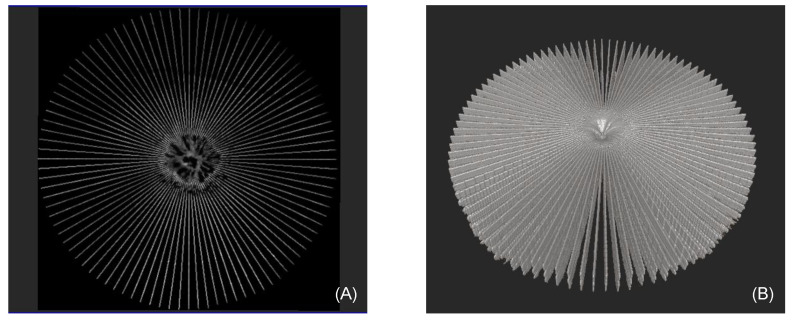
(**A**) shows a coronal and (**B**) an aerial view of the same full-thickness macular hole, based on the reconstruction of the radial volumetric OCT at a level of approximately 40 µm above the retinal pigment epithelium (RPE), which corresponds to the photoreceptor layer (ellipsoid zone, EZ).

**Figure 8 sensors-22-01135-f008:**
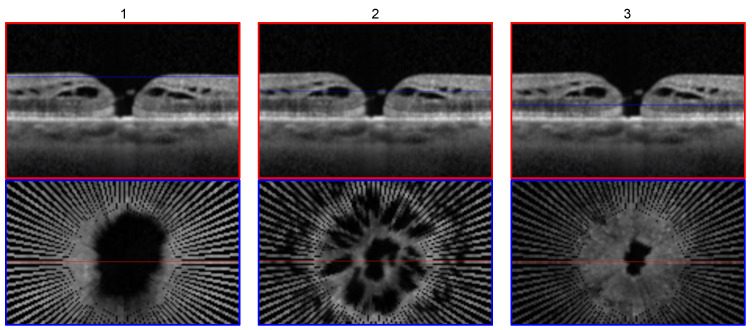
Close-up view of a volumetric reconstructed macular hole surrounding the volumetric center at three different levels (row 1; blue line). Row 2 shows the corresponding top view/coronal height cut to each image in row 1. The levels shown are just below the internal limiting membrane (ILM) in column 1, inner plexiform layer (IPL) in column 2 and at the height of the minimum linear diameter (MLD) in column 3.

**Figure 9 sensors-22-01135-f009:**
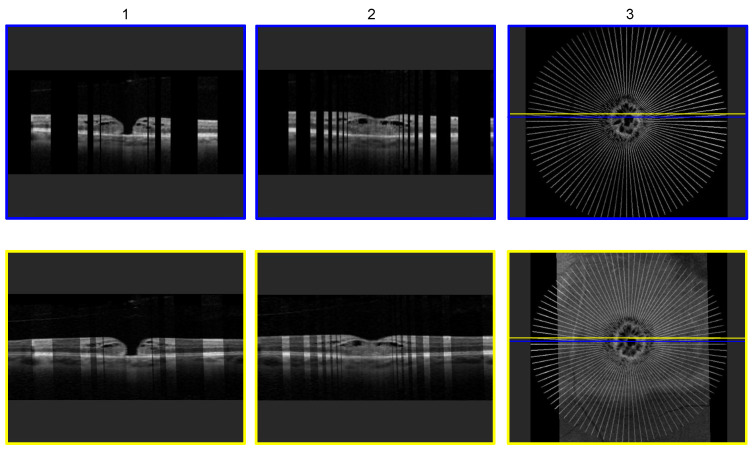
Comparison of an ordinary 48 radial volumetric OCT reconstruction (row 1) and a radial and linear fused volumetric OCT reconstruction (row 2). Two different, randomly chosen, cross-sectional B-scan slices were chosen (column 1 and column 2). In row 1, column 1 and column 2, the dark areas represent missing information. In row 2, column 1 and column 2, the brighter areas represent information provided by the radial volumetric OCT and the darker areas information provided by the linear volumetric OCT. The last column shows the top view of each reconstruction technique corresponding to the images in the same row, with the colored lines indicating the location of the cross-section from which the B-scan slice originates.

**Figure 10 sensors-22-01135-f010:**
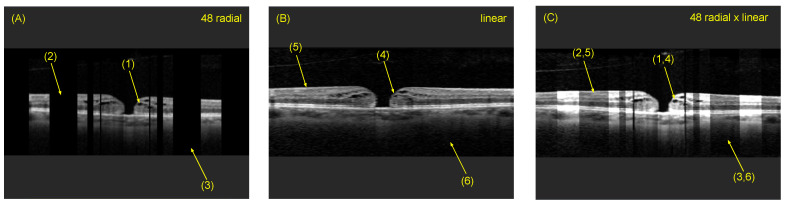
Visualized example of the information gain by radial and linear reconstruction. Image (**A**) shows a B-scan slice taken from the 48 radial B-scan slice reconstruction and Image (**B**) shows the same cross-sectional B-scan slice from the linear reconstruction. Image (**C**) shows the resulting fused OCT after selective masking of the linear OCT (**B**), compounding it with Image (**A**). At location (2,5), (3,6) and (1,4), the information gain is clearly visible (different intensities represent different reconstructions).

**Figure 11 sensors-22-01135-f011:**
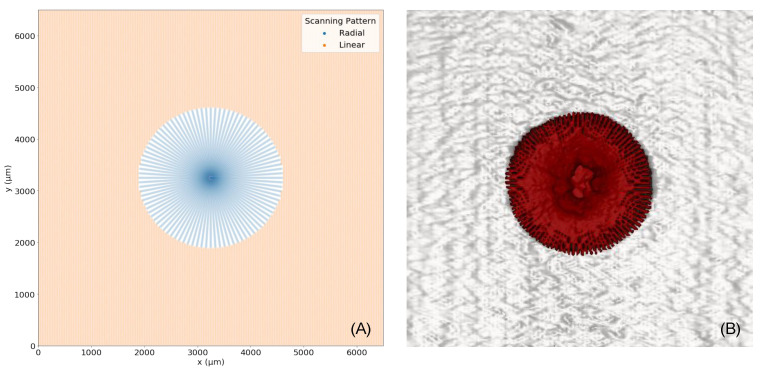
Radial and linear A-scans until a certain radius R. (**A**) represents the simulated fusion of both scanning patterns. (**B**) represents the real fusion of both scanning patterns.

**Figure 12 sensors-22-01135-f012:**
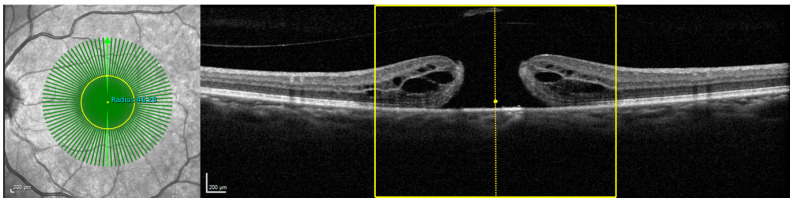
The area within which the radial scanning acquisition has higher A-scan density than the linear scanning acquisition, projected on a B-scan slice, is highlighted. On the fundus image, this area is presented as a circle.

## Data Availability

The patient data presented in this study are available on request from the corresponding authors. The data are not publicly available due to privacy.

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
