# Peer review of "A Tool for High-Resolution Volumetric Optical Coherence Tomography by Compounding Radial-and Linear Acquired B-Scans Using Registration"

_sensors, 2022, doi:10.3390/s22031135_

Round 1
Reviewer 1 Report
This paper(no.1558284) describes the method to realize the high resolution of OCT by compounding the radial scanned image and the B-scanned image. In the introduction, the back grounds and needs for enhancements of resolution and objects of this study are written mainly in the field of ophthalmology. In the methods, the proposed method in this paper are explained step by step, such as linear volumetric OCT, radial volumetric OCT, iterative closest points, and compounding. In the results, radial scanned OCT, linear scanned OCT, and compounded OCT are compared to show the validity of the proposed method.
In practical clinical applications of OCT to ophthalmology, it is very significant to increase the spatial resolution of OCT around the macular hole. Showing practical OCT images of eyes, the level of this paper reached at this Journal of “Sensors.” However, considering that this Journal is technical, not clinical, so technical story must be clear. Therefore, I recommend that the paper will be published from this paper after revisions mentioned below.
1.“High-resolution” in the title is key word. So in the story, firstly the resolution is defined or explained, second the proposed method is explained, third the practical enhancement of resolution should be discussed by proposed compound method.
However, at present the resolution is not described clearly, and its enhancement is also not clear. For example, in the “2.5 Enhancement of spatial resolution”, the spatial resolution by proposed compound method will be defined and explained. It concerned with the density of scanned lines and also concerned with the distance from the center.
- p.3, L.91
The specifications of OCT system should be described, especially about the wavelength, the depth and lateral resolutions.
- There are no descriptions in text about Fig.3, Fig.4, Fig.11, and Fig.12.
- p.9, L.219
In Fig.9 (a) radial scanned OCT, there are not image information in a few regions (2) and (3). Please describe the reasons.
- p.9, L.233
I feel that “3.5 Inflection point” is explanations about inflection point by itself, and should be moved into “method”.
- p.11, L.264
Because there are movements in tissues, the relation of spatial resolution and the measurement time should be considered and discussed. If you realize the high resolution, you must measure the OCT images in a short time. In this study, it takes the time to measure radial scanned, B-scanned, and processing. I think it is important point in the practical applications.
END
Author Response
Dear reviewer,
Please see the attachment.
Kind regards,
Christian Bosch

Reviewer 2 Report
The manuscript by Christian M. Bosch et al. demonstrated optical coherence tomography with linear and radial acquisition patterns. A technique combining the advantages of high-density radial and linear B-scans was proposed to enhance the image quality and usability of volumetric OCT. Besides, the author also employed a method known as the iterative closest point (ICP) variant to register both volumes. Overall, the results of this study should be of interest to the biomedical optics community. Specific comments are as follows:
- In line 100, because 360/3.75 should be equal to 96, please check again how to calculate 48 individual B-scans?
- In equation (3), please add and describe the meaning of V and U. Also, what is the relationship between t in equation (4) and T in equation (2)?
- In line 246, please add the unit (i.e., degree or um) for the value 1.5. Also, how did the authors calculate this value of 1.5?
- Which software tool did the author use when making the OCT post-processing and image registration? Also, how long does this calculation for ICP take?
- In column 3 of fig. 8, what is the meaning of the color of two outer frames?
- When making the acquisition of OCT images with linear and radial acquisition patterns, did the authors encounter severe motion artifacts? Also, how did the authors deal with this issue?
Author Response

(The authors gave the same response as above.)

Round 2
Reviewer 1 Report
No comments